# A New Treatment Protocol of Combined High-Dose Levothyroxine and Repetitive Transcranial Magnetic Stimulation for the Treatment of Rapid-Cycling Bipolar Spectrum Disorders: A Cohort Evaluation of 55 Patients

**DOI:** 10.3390/jcm11195830

**Published:** 2022-09-30

**Authors:** Antonis C. Zamar, Christos Kouimtsidis, Abbi Lulsegged, Robin Roberts, Theodoros Koutsomitros, Daniel Stahl

**Affiliations:** 1The London Psychiatry Centre, London W1G 7HG, UK; 2Imperial College, London SW7 2BX, UK; 3Health121 Ltd., London EC4M 7JN, UK; 4Aupres Medical, 1 Harley Street, London W1G 9QD, UK; 5Greek rTMS Clinic, Medical Psychotherapeutic Centre (ΙΨΚ), 546 24 Thessaloniki, Greece; 6Department of Cognitive Neuroscience, Faculty of Psychology and Neuroscience, Maastricht University, 6211 LK Maastricht, The Netherlands; 7Statistics Kings College, Strand, London WC2R 2LS, UK

**Keywords:** bipolar disorder, treatment, levothyroxine, repetitive transcranial magnetic stimulation (rTMS), thyroid test

## Abstract

Background: Bipolar spectrum disorders (BSD) are highly disabling, with rapid cycling being treatment resistant. High-dose levothyroxine (HDT) has been reported to be effective. Diagnosis is associated with mutations in thyroid-activating enzymes and cerebral transporter protein carrier. Repetitive transcranial magnetic stimulation (rTMS) has neuroplastic effects. Methods: We report data on 55 severely symptomatic patients with rapid-cycling BSD treated with a combination protocol of HDT and rTMS. Of the patients, 31 patients (56.4%) were female and 40 (72.7%) had at least one additional diagnosis. Results: Patients were evaluated at three monthly intervals after acute treatment. Remission was measured using the Sheehan Disability Scale (SDS). The average number of medications prescribed was 1.8, with 32 patients (58.2%) needing only levothyroxine. The average dose of levothyroxine was 303.7 mcg (50 mcg–1000 mcg). A total of 53 patients were in remission (96.4%), with an average duration of 2.0 years. The SDS scores decreased significantly (Cohen’s d = 2.61 (95% C.I. 1.81 to 2.83, *p* < 0.001). One patient had reversible side effects. A total of 52 (94.3%) patients had Deiodinase 1 and 2 (DiO1/DiO2) or SLCO1C1 protein carrier gene mutations. Conclusion: The data support the safety and acceptability of combined HDT/rTMS. Patients achieved long remissions with substantial improvements in quality of life.

## 1. Introduction

Bipolar disorder (BD) is a common and disabling mental illness with significant morbidity and mortality [1,2]. The understanding of BD is evolving over the years with changes to the definition and criteria for classification. BD I is characterised by a minimum of 7 days of mania. BD II is characterised by at least 4 days of hypomania. BD not-otherwise specified (BD-NOS) (as per DSM 5 296.8) or unspecified (as per ICD-10 F31.9), or subthreshold BD, is characterised by any form of bipolarity which does not fit BD I and BD II, including the presence of at least one manic symptom in the presence of a depressive episode, such as flight of ideas, irritability, impulsivity, or high libido. BD-NOS is the most common presentation, with a 1.4% incidence versus 0.6% in BP I and 0.4% in BP II [1]. In the United States, reported prevalence for BD-NOS is 2.4%, that for BD I is 1.1%, and that for BD II is 1% [3]. The term bipolar spectrum disorders (BPS) has been proposed to include all forms of disorder in a continuum that helps both research and clinical management [4]. Rapid-cycling BD is a pattern of frequent, distinct episodes in bipolar disorder. It is associated with a more malignant course and is even more difficult to treat. Cycling can be rapid over weeks, ultrarapid over days, or even ultradian, with changes of polarity within the day, which may well be the subthreshold presentation by definition [5].

In addition, BP I and II are plagued with subthreshold symptoms, which hamper recovery, affect cognition and quality of life [6], and increase relapse rates and mortality [7]. Furthermore, these symptoms failed to reach remission with quetiapine XR in addition to mood stabilisers with failure to respond/remit in subthreshold manic symptoms and failure of remission in depressive symptoms, posing a serious challenge in treatment [8].

Alongside the evolution of the understanding of BPS, treatment guidelines have been changed. Whereas there is an overall agreement across guidelines regarding the treatment of BD I and BD II, there is far less certainty regarding subthreshold presentations. Existing treatment for BD I and II is less effective and would require an average of 3.8 medications [9].

Transcranial magnetic stimulation (TMS) is a non-invasive treatment for depressive episodes of bipolar disorder by using magnetic pulses to depolarize cerebral neurons. Systematic review and meta-analysis (Evidence level 1, recommendation grade A) of rTMS for bipolar depression and BD indicated that it is safe and effective with low risk of side effects [10,11,12]. High-dose levothyroxine (HDT) is shown to be well-tolerated in BPS and effective even in doses of 150 to 1000 mcg [13,14,15]. Given the known influence of genetic factors in both BPS and circulating thyroid hormone (TH) it could be hypothesised that there might be common genetic factors that could explain both the aetiology of BPS and effectiveness of HDT in BPS as well as the lack of expected side effects of HDT [16]. The present authors have reported data in the first 20 patients with BPS treated and genetically tested consecutively [13]. Nineteen showed single nucleotide polymorphism (SNPs) in Deiodinase 1 (DiO1) and Deiodinase 2 (DiO2), and one showed SNPs in the SLCO1C1 protein carrier gene, which is associated with the intracellular transport and availability of thyroid hormones in the brain tissues as well as the periphery [16]. DiO1 is responsible for the activation of thyroxine (T4), the thyroid prohormone to triiodothyronine (T3), and the active thyroid hormone in the liver and kidney (periphery), whilst DiO2 is responsible for the same conversion in the brain and placenta [17,18]. Furthermore, a change in T4/T3 ratio from 4:1 pre-treatment to 5:1 post treatment, with T4 markedly elevated but T3 largely normal, was observed. A high T4/T3 ratio at day 10 since the start of HDT treatment predicted good tolerability of HDT [13].

In this study, we investigate the combined effect of prescribing high doses of levothyroxine and transcranial magnetic stimulation (TMS) in clinical practice. The aim of the current study is to assess (i) the safety, tolerability, acceptability of the combined treatment protocol; (ii) any impact on the quality of life; (iii) the presence of gene polymorphism; and (iv) thyroid hormones levels. 

## 2. Materials and Methods

This report is of two separate treatments recommended for the treatment of the 13th edition of the Maudsley prescribing guidelines [19], but given the reported lack of side effects and patient acceptability, these treatments were used in combination. Given this is a day-to-day practice matter of treatments within guidelines, LREC approval was not required. All patients were given a choice of treatment based on the Maudsley prescribing guidelines 2018, 13th edition [19], with pros and cons of each rapid cycling treatment discussed. Studies on HDT were reviewed with patients, including the Pilhatsch et al. [14] and Waslshaw et al. [15] RCTs. Patients were given the choice, and depending on preference whilst awaiting genetic results, rTMS or Levothyroxine was started first (cost decision) or both were started simultaneously.

We report on the treatment of a cohort of patients with BPS treated at a London outpatient private facility in the UK for a rapid-cycling presentation with predominant severe depressive, mixed, hypomanic, and depressive cycles (see below in Section 3). They were all treated with the new protocol of combined HDT/rTMS. rTMS protocols used were right-sided low frequency (1 HZ) stimulation for one week followed by high-frequency right side (20 Hz) stimulation. Depending on clinical presentation, the low-frequency stimulation was extended beyond the first week if anxiety was prominent.

Blood tests to exclude organic mood disorder were undertaken, including thyroid function tests (TFTs), full blood count (FBC), liver function tests (LFTs), calcium, urea and electrolytes, ESR, ferritin, glucose, and lipids at the outset and then TFTs at day 10, including reverse T3 (rT3) at 150 mcg, 400 mcgs, 500 mcgs, and at final remission dose. The aim of TFT checks was to check that thyroid doses were supraphysiologically suppressing TSH with a minimum double free T4 to check for absorption and compliance [20].

The Lifecode Gx™ central nervous system panel was tested, which—apart from dopaminergic, adrenergic, serotoninergic, and other central nervous system (CNS) neurotransmitters—specifically tests for the rs2235544 DiO1 variant, 2 variants of DiO2—namely rs12885300 Gly3Asp and rs225014 Thr92Ala—as well as the SLCO1C1 solute carrier family 21, member 1C1, and rs10770704 intron3C>T variant.

Patients were asked to discontinue all substances affecting mitochondrial function, namely caffeine—which is known to affect cortisol levels [21], which in turn affect mitochondria [22]—and alcohol [23]. Levothyroxine was started at 50 mcgs and according to a recognised protocol [20] increased by 50 mcg every 4 days up to 150 mcg. After 2–3 days, 150 mcg blood tests looking at fT4, fT3, TSH, and rT3 were conducted to assess absorption, T4/T3 ratio and rT3 levels, and ECGs were conducted. Doses were increased by 50 mcg per week thereafter depending on tolerability and progress. The aim of supraphysiological doses of levothyroxine is to supress TSH and achieved a free T4 level of at least double the baseline [20]. Dose increases were stopped at recovery and maintained throughout remission. If symptoms returned doses were increased or rTMS reintroduced until a 4-month period of stability was achieved. The SDS was measured at the outset, recovery, and final follow-up during the period of data collection. This was correlated with patients’ reports in the notes and upon review, as well as relatives′ comments when available.

Data were collected during the period of March–September 2021, at the regular three months review clinical appointment. Duration of remission was verified from patient’s case notes and refers to the most recent to the review period. Other changes reported refer to changes between start of HDT treatment and review, with additional report of change at day 10 since start of HDT treatment for thyroid hormones levels. Severity of the illness (including side effects of treatment) was measured by its impact on overall functioning using the Sheehan Disability Scale (SDS) (primary outcome), which is a self-report measure [24]. Patients are asked to rate how their symptoms have disrupted their work, social life, and family life. Each domain is rated on a scale from 0–10, and the average across the three domains is reported. Domain scores greater than 5 suggest significant impairment in that domain, and a score of 7 or above suggests severe/very severe impairment [24]. The scale is very brief and simple to use. SDS has demonstrated sensitivity to the effects of treatment [25]. Descriptive statistics are reported as mean, median, and upper and lower range. We performed a *t*-test to test for differences between two groups and mixed models to assess changes of hormone levels over time. For the mixed model analyses, we used an unstructured covariance matrix of residuals to control for repeated observations over time.

## 3. Results

### 3.1. Socio-Demographic and Clinical Characteristics

The cohort consisted of 55 patients, with 31 (56.4%) self-identified as female, mean age at presentation at the treatment centre of 37.3 years (median: 40, range: 18–69), and age of onset 21.5 years (median: 18, range: 7–44). All patients were booked to receive rTMS, but one was discharged before starting to receive rTMS. All patients were diagnosed with BSD (BD NOS: 47, BD I: 4 and BD II: 5). Forty patients (72.7%) had at least one additional diagnosis, with the most prominent being PTSD (19, 34.5%), ADHD (16, 29.1%), and GAD (5, 9, 1%). 

### 3.2. Tolerability

Mean duration of treatment with current protocol was 4.3 years (median: 3, range: 0–13). Only one patient had side effects, which consisted of feeling hot and palpitations. Symptoms subsided when the dose was stopped and did not reappear when the dose was re-titrated. The remaining patients reported no side effects. No patient had to discontinue treatment. 

### 3.3. Primary Outcome

Figure 1 show the changes of SDS from start of treatment to at time of review. The SDS score was reduced in 55 patients from 7.33 (median: 8; range: 1.4–10) at the start of treatment to 1.27 (median: 0.3; range: 0–8.6) at the time of review, which was significant (mean change 6.06 (95% C.I. 5.36 to 6.77), Cohen’s d = 2.61 (95% C.I. 1.81 to 2.83), t(54) = 17.23, *p* < 0.001). While no patient scored 0 initially, eighteen patients (32.7%) scored 0 at review; 48 patients scored above the clinical cutoff 5 initially, with only 3 scoring above 5 at review.

### 3.4. Secondary Clinical Outcomes

At the time of the evaluation reported here, 53 patients were in remission (96.4%), with an average duration of 2.0 years (median: 1.5; range: 0.2–6). The mean duration of time to reach initial remission was 42.6 weeks (median: 10.5; range: 1–737) and the average dose of levothyroxine was 303.7 mcg (median: 300; range: 50–600). Two patients never reached remission; one was discharged before starting rTMS treatment, and one was not compliant with HDT. Five patients following remission have relapsed during the period of treatment. For three of them, there was no precipitating factor identified; two were non-compliant with monthly maintenance rTMS or HDT; and two have initially improved on low dosages of Levothyroxine, relapsed, but then recovered on higher doses. 

The average dose of levothyroxine at review was 423.2 mcg (median: 400; range: 50–1000). For 32 patients (58.2%), levothyroxine was the only medication prescribed. The average number of medications prescribed was 1.8 (median: 1, range: 1–6) including anti-anxiety (19, 34.5%), antipsychotics (12, 21.8%), sleep medication (13, 23.6%), mood stabiliser/Lithium (4, 7.3%), ADHD (2, 3.8%), and other medications (4, 7.6%). No antidepressants were prescribed. 

A total of 54 patients received genetic testing at the start or early stages of treatment, for two patients, data for two mutations were only available. Of 52 patients with complete data, 50 (94.3%) were identified as having mutations on DiO1, DiO2, or SLCO1C1. The number of mutations did not correlate with a change in SDS (r = 0.08, *p* = 0.56, *n* = 52), and there was no significant difference in mean number of mutations between patients who relapsed (mean = 3.3) and those who did not relapse (mean = 3.6, t(50) = −0.29, *p* = 0.77). 

### 3.5. Thyroid Hormone Levels

Figure 2 shows the changes of T3, T4, and T4/T3 levels over time. Mixed-model analyses showed that mean T3 levels slightly but non-significantly increased from 4.9 (median 4.60, range: 2.1–12.4) at treatment start to 5.5 (median: 5.4, range: 3.2–8.9) at day 10 of treatment (*p* = 0.38), and increased significantly to 10.6 (median: 8.8, range: 3.7–29.0) at the review time point (*p* < 0.001). Mean T4 levels increased significantly from 16.3 (median:16.2, range: 6.0–26.7) at treatment start to 25.0 (median: 25.4, range 4.7–36.0) at day 10 treatment (*p* = 0.005) and increased significantly to 53.0 (median: 50.4, range: 17.2–100.0) at the review time (*p* < 0.001). The changes in T3 and T4 levels resulted in a significant increase in T4/T3 ratio from 3.7 (median: 3.5, range: 0.5–8.8) at treatment start to 4.5 (median: 4.8, range: 1.0–6.0) at day 10 of treatment (*p* = 0.005) and a reduction to 3.5 (median: 3.3, range: 0.9–12.7) at the review time point (*p* = 0.001), reaching a similar ratio as at baseline (*p* = 0.62). 

## 4. Discussion

Data reported here support observations reported by the same authors on a previous cohort of 20 patients [13] and suggest that the combination of HDT and rTMS is safe, acceptable, and well tolerated by patients, leads to prolonged period of remission and require combination of less medication as opposed to existing treatment, hence reducing the risk of interactions and side effects. The same mutations were observed as in previous reported evaluation of treatment and similar increase of hormone ratio of T4/T3. It is of great importance to note the safety of the combined protocol. This was despite taking very high doses of Levothyroxine and having blood results typically seen in the thyrotoxic range. Similar results are reported in patients with refractory depression treated with HDT up to 500 mcg once daily [26]. The results could be explained by the actions of DiO1 and DiO2 (see below). 

Normal thyroid hormone homeostasis involves secretion of thyrotropin releasing hormone from the hypothalamus which stimulates the pituitary gland to produce Thyroid Stimulating Hormone (TSH). TSH promotes release of thyroid hormones, thyroxine or T4 and T3. T3 is the most active form of thyroid hormones. Approximately 80% of T3 is derived from peripheral conversion of T4 to T3 and 20% from direct release from the thyroid gland. Two deiodinase enzymes, deiodinase iodothyronine type I (DiO1) and deiodinase iodothyronine type II (DiO2), are responsible for conversion of T4 to T3. DiO1 converts T4 to T3 principally in the circulation, while DiO2 is responsible for intracellular conversion of T4 to T3. Conversely, DiO1 and deiodinase iodothyronine type III (DiO3) inactivate T4 and T3, converting it to reverse T3 and T2 respectively. The latter helps to protect the body from the effects of too many thyroid hormones. 

Genetic studies have shown that polymorphism of the DiO2 gene (rs225014; T92A) is associated with depression, and heterozygote polymorphism has been associated with an increased risk of bipolar disorder [27]. 

It is not clear how SNPs of DiO2 might be associated with BD, but the variant form of DiO2 has been shown in molecular studies involving post-mortem cerebral cortex cells to accumulate in the Golgi body rather than be recycled efficiently to the endoplasmic reticulum; this results in impaired conversion of T4 to T3 in the endoplasmic reticulum and increased endoplasmic reticulum stress [28]. 

Animal studies have previously shown that T4 is an important source of intracellular T3 in the brain and this might help to provide one possible explanation for why high doses of T4 might help in Bipolar disease [29].

The physiological mechanism associated with the potential effectiveness of the combination of HDT and rTMS is not known. However, it might be associated with intracellular changes via mitochondriogenesis [30,31,32] and increased cerebral blood flow and oxygen utilization [33] associated with the combination of the effect of each of the treatment components. 

The effectiveness reported by the present authors [13] in previous and current cohorts might suggest a state of “cerebral hypothyroidism” due to DiO1, DiO2, and/or SLCO1C1-protein-transporter-encoding gene mutations, which regulate levels of thyroid in the brain tissues, which have been corrected by HDT. In turn, impaired thyroid metabolism may well be responsible for the mitochondrial dysfunction identified in BPS. The abnormal mitochondrial function would in turn influence neuronal function, including neuroplasticity, needed following multiple episodes and rapid cycling, hence creating a vicious cycle [28]. We hypothesize that the target for treatment of rapid-cycling bipolar disorders should be the mitochondria, given that rTMS-induced neuroplasticity is mitochondria-dependent [22] and thyroid hormones exert a significant impact on mitochondrial function, thus enhancing neuroplasticity. In addition, one should consider the use of supplements or additional interventions supporting mitochondrial function in the treatment of rapid cycling.

This study was a real-life cohort study (CEBM evidence level 2B) and added to the previous 20 patient cohort (2B), hence giving a Grade B recommendation for the use of HDT/rTMS in terms of effectiveness. There are no Grade A recommendation studies for other treatments and certainly none to our knowledge for subthreshold bipolar disorders. With regards to safety, both HDT and rTMS are Grade A recommendations with consistent level 1 studies for safety. Further well-designed randomised controlled trials are required to establish a higher-level evidence base for the combined HDT/rTMS treatment, but rapid cycling poses significant challenges as end points may not be as informative as longitudinal evaluations using mood charts as per the Walshaw et al. study [15].

## 5. Conclusions

Rapid-cycling bipolar disorders are conditions with very high mortality and morbidity rates. Data reported here support observations reported by the same authors on a previous cohort of 20 patients [13] and suggest that the combination of HDT and rTMS is safe, acceptable, and well tolerated by patients. The combination leads to a prolonged period of remission and requires less medication as opposed to existing treatment, hence reducing the risk of interactions and side effects. The same mutations were observed as in previous reported evaluation of treatment and similar increase in hormone ratio of T4/T3 [13]. Further research is required on a preclinical level to understand both how the treatment works on physiological and genetic levels as well as to establish treatment effectiveness and cost-effectiveness with a randomised controlled trial.

## Figures and Tables

**Figure 1 jcm-11-05830-f001:**
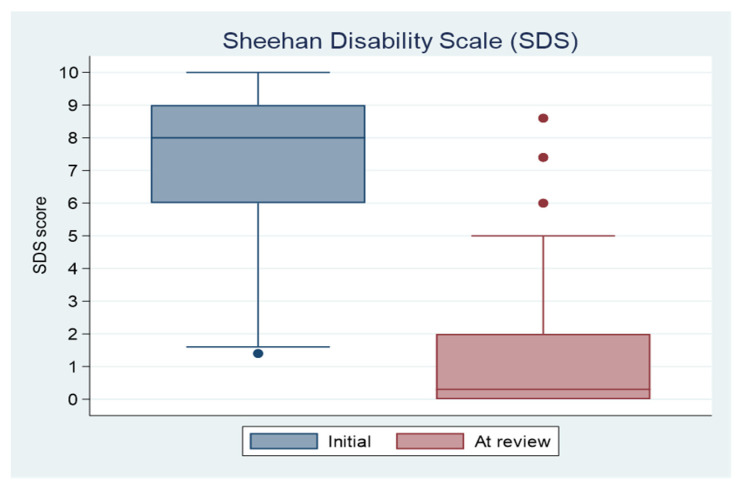
Box plots of severity of dependence scale (SDS) at start of treatment and at time of review. The central mark of the box shows the median value of the scores and the lower and upper top edges of the box indicate the 25th and 75th percentiles, respectively. Outliers are designated with a dot. The whiskers show the most extreme data points not considered outliers.

**Figure 2 jcm-11-05830-f002:**
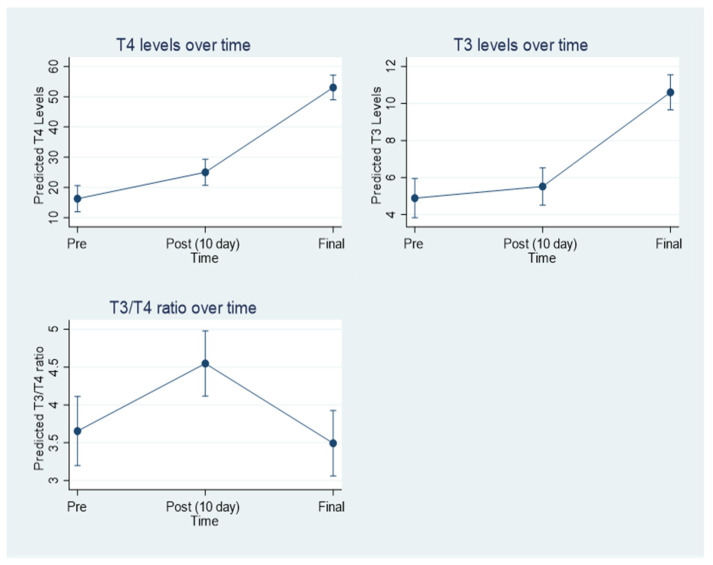
Shows the marginal means of T3 and T4 levels and T4/T3 ratio of the multilevel modelling analyses for pre, post, and final observation times.

## Data Availability

Data available on request due to restrictions. The data presented in this study are available on request from the corresponding author.

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
