# Peer review of "A New Treatment Protocol of Combined High-Dose Levothyroxine and Repetitive Transcranial Magnetic Stimulation for the Treatment of Rapid-Cycling Bipolar Spectrum Disorders: A Cohort Evaluation of 55 Patients"

_jcm, 2022, doi:10.3390/jcm11195830_

Round 1
Reviewer 1 Report
In the Abstract:
It is unclear if the entire cohort received high dose thyroid hormone and rTMS or monotheyrapy with levothyroxine
In general:
-Are you referring to treating the depressive phase of bipolar spectrum disorder, mixed phase, or any phase? Please clarify and be specific.
-Abbreviations should be elaborated the first time they are used (SNP DiO1 DiO2 SLCO1C1 T4 T3)
In the methods:
-Please clarify the rTMS protocol you are using for the patients
-Please list the entire genetic panel you are testing
-Please list the blood work and frequency
-Please list the time line of the two treatments (are they simultaneous or do you first give levothyroxine and then rTMS
-In which patients do you decide to give Liothyronine?
In the discussion
-Please comment on other others dosing guides and efficacy of high dose thyroid for bipolar
-Please comment on other studies of rTMS for bipolar
-Please comment on limitations and next steps
Reviewer 2 Report
The paper has an interesting focus on subthreshold presentations of bipolar disorders, which are meaningful because such symptoms prevent recovery and foster relapses. Moreover, this is a very interesting sample. The transferability to everyday clinical practice should be given.
Unfortunately, based on some identified weaknesses, I cannot recommend the manuscript for acceptance:
Introduction:
A clear definition of bipolar disorder as distinct from the other presentations would be appreciated.
Is reference 5 (“The efficacy of repetitive transcranial magnetic stimulation (rTMS) for bipolar depression”) really related to the sentence: “High dose levothyroxine (HDT) 50 is shown to be well tolerated in BPS and effective (5)”?
Line 62: “The physiological mechanism associated with the potential effectiveness of the combination of HDT and rTMS is not known. It might be associated though with intracellular changes with mitochondriogenesis [11] and increase Cerebral Blood Flow and oxygen utilization [12], associated with the combination of the effect of each of the treatment components. BPS are hypothesised to be associated with mitochondrial dysfunction [13,14,15].”
Although the exact mechanisms of action of the combination of both therapies are unclear, more could be said about putative mechanisms of each intervention (TMS and HDT), e.g., serotonergic modulation or induction of neurogenesis.
Methods:
No ethics approval is given. The authors should either submit the ethics approval later or briefly explain why it may not be required for the study.
Important details of the rTMS intervention, such as localization, intensity, frequency are not explained or referenced anywhere.
Results:
The section consists only of subheading „3.1. Tolerability“. However, there should be other subheadings like “sample, effectiveness”…
The manuscript contains only one figure. One would expect it to contain the core message. However, the present graph contains only the rough time courses of peripheral thyroid hormone levels. Why these courses are of such great interest is not adequately explained either in the introduction or in the discussion.
Discussion:
The discussion is extremely brief and does not account for the complexity of the various topics: rTMS, HDT, genetic markers. A discussion of the findings in the light of the current data is missing.
Conclusion:
Line 158: “Rapid cycling bipolar disorders are conditions with very high mortality and morbidity rates.”
The subject of "rapid cycling" has not been mentioned anywhere in the manuscript so far. Why then does it suddenly appear in the conclusions?
In summary, in my view, the manuscript in its current form is not suitable for acceptance in a leading journal such as JCM and needs a fundamental reorganization. The complex topics could be better addressed in a longer format (not as a short report as here). However, I find the topic, the sample and the methods very exciting. I hope that my comments will be taken as constructive criticism.
